# Piezoelectric Sensor for the Monitoring of Arterial Pulse Wave: Detection of Arrhythmia Occurring in PAC/PVC Patients

**DOI:** 10.3390/s21206915

**Published:** 2021-10-19

**Authors:** Cheng-Yan Guo, Kuan-Jen Wang, Tung-Li Hsieh

**Affiliations:** 1College of Medicine, National Taiwan University, Taipei 10617, Taiwan; r04458006@ntu.edu.tw; 2Accurate Meditech Inc., New Taipei City 24159, Taiwan; stanley@accurate-meditech.com; 3General Education Center, Ursuline College of Liberal Arts Education, Wenzao Ursuline University of Languages, Kaohsiung 80793, Taiwan

**Keywords:** piezoelectric, Moens–Korteweg equation, PAC, PVC

## Abstract

Previous studies have found that the non-invasive blood pressure measurement method based on the oscillometric method is inaccurate when an arrhythmia occurs. Therefore, we propose a high-sensitivity pulse sensor that can measure the hemodynamic characteristics of the pulse wave and then estimate the blood pressure. When an arrhythmia occurs, the hemodynamics of the pulse wave are abnormal and change the morphology of the pulse wave. Our proposed sensor can measure the occurrence of ectopic beats from the radial artery, and the detection algorithm can reduce the error of blood pressure estimation caused by the distortion of ectopic beats that occurs when the pulse wave is measured. In this study, we tested patients with premature atrial contraction (PAC) or premature ventricular contraction (PVC) and analyzed the morphology of the pulse waves when the sensor detected the ectopic beats. We discuss the advantages of using the Moens–Korteweg equation to estimate the blood pressure of patients with arrhythmia, which is different from the oscillometric method. Our research provides a possible arrhythmia detection method for wearable devices and can accurately estimate blood pressure in a non-invasive way during an arrhythmia.

## 1. Introduction

In previous studies, arrhythmia has affected the accuracy of blood pressure measured by an oscillometric-based electronic sphygmomanometer [1]. The blood pressure measurement function is attached to a wearable device with non-invasive blood pressure measurement technology. It is necessary to overcome the inaccuracy of the oscillometric method in estimating blood pressure when an arrhythmia occurs. Electronic sphygmomanometers currently on the market, such as the Microlife B3 AFIB, can also detect atrial fibrillation (AF) belonging to arrhythmia [2]. When an arrhythmia occurs, the strength of the pulse wave of the peripheral artery changes [3,4], and the arrhythmia can be detected by monitoring the R–R interval of each pulse wave signal in the envelope curve [5]. The current electronic sphygmomanometer can detect an arrhythmia, but the blood pressure measurement result is inaccurate because the arrhythmia causes the hemodynamics of the artery to change [6]. The pulse wave amplitude attenuates, causing distortion, so the corresponding systolic and diastolic pressure points cannot be accurately estimated from an envelope curve measured with the oscillometric method.

The piezoelectric sensor we propose can avoid the distortion of the pulse wave measured by the oscillometric method when an arrhythmia occurs and detect and filter out arrhythmia through a dynamic threshold algorithm. Two piezoelectric sensors are arranged to measure the pulse wave velocity (PWV) of the radial artery, and the blood pressure is estimated by the Moens–Korteweg (MK) equation based on the hemodynamics [7]. This enables the wearable device to measure the blood pressure of patients. PAC) and premature ventricular contraction (PVC) are the most common types of ectopic beats that can mimic the irregular beat pattern typical of AF [8,9]. Therefore, this study aims to detect the pulse wave of PAC/PVC patients and assess the influence of PAC/PVC on PWV.

Currently, several research methods for measuring the PWV of local arteries include the direct use of two pressure cuffs to pressurize between the upper arm and the wrist to obtain the PWV of the two-point pulse wave [10]. However, the size of such a device is too large to be suitable for a wearable device. Two or more photoplethysmography (PPG) probes can also measure the pulse wave of the local artery [11]. However, the diffuse reflectance spectrum (DRS) pulse wave reflects the change of hemoglobin and oxygenated hemoglobin in the artery, and a correction of the regression model needs to be applied based on blood flow because that change is not a mechanical wave. Multiple sets of bio-impedance-based sensors can also be deployed. When the artery expands due to pulse waves, the bio-impedance will change. PWV can be obtained by measuring the bio-impedance changes at two points in the local artery. A bio-impedance sensor needs two parts: AC signal output and signal receiver [12,13]. The position where the AC signal is applied on human tissue is not easy to determine. Due to the limited length of wearable devices, the distance between measurement points is prone to errors. The measurement method based on a motion sensor, with the placement of dual accelerometer probes over the artery [14], measures the pulse wave by the local artery. In the application scenario of wearable devices, the accelerometer needs to eliminate the interference caused by the user’s movement.

Therefore, to accurately measure the PWV of the local artery, the sensor needs to have good rigidity. The relative position of the two pulse waves remains unchanged at any time. It is sensitive to vibration signals and can sense when an arrhythmia occurs. The piezoelectric film is a sensor made of polyvinylidene fluoride (PVDF). It has thinness and flexibility and can integrate into wearable devices [15]. The cost of piezoelectric film is usually in the tens of dollars. We have considered the balance between cost and measurement performance. The pulse wave sensor designed in this research is a piezoelectric ceramic based on a buzzer, with the advantages of being easy to get and low cost. The manufacturing process of plastic parts encapsulating piezoelectric ceramics can be completed using a 3D printer without the need for a micro-electro-mechanical systems (MEMS) process [16], making it easy to produce a pulse wave sensor that can detect the occurrence of arrhythmia.

### Our Contribution

The contribution of this research is the design of a small, low-cost, and highly sensitive pulse wave sensor that can measure the pulse wave signal when an arrhythmia occurs. In addition, we designed an algorithm based on the dynamic threshold to detect the pulse wave of the PVC/PAC waveform. Our method avoids the measurement error caused by the pulse wave distortion due to arrhythmia and also uses the MK equation based on hemodynamics to estimate blood pressure. Finally, we verified that subjects with arrhythmia with PAC/PVC could accurately estimate their individual blood pressure.

## 2. Materials and Designs

Piezoelectric ceramics can sense tiny mechanical waves. Pressure waves are a kind of mechanical wave that can deform piezoelectric ceramics. When the pressure generated by the pulse wave transmitted by the radial artery deforms the piezoelectric ceramics, the piezoelectric ceramics can convert the pulse wave into a voltage signal. This chapter explains the cavity structure of the small piezoelectric sensor we designed. This cavity structure enables the piezoelectric sensor to be integrated into the wearable device. The pressure of the pulse wave is sensed when the device is close to the radial artery. Air pressure will be concentrated on piezoelectric ceramics, ensuring that the received pulse wave signal has a better signal-to-noise ratio.

### 2.1. Mechanical Design

Figure 1 shows the cavity structure of the pulse wave sensor proposed in this study. The piezoelectric ceramic is fixed between the top and bottom covers, and the joint between the covers is filled with a sealing gel to eliminate any gap around the piezoelectric ceramics and dissipate the pressure of the pulse wave. In addition, the piezoelectric sensor has a rubber cap on its side to measure the pulse wave. The rubber cap allows the piezoelectric sensor to adsorb on the human skin easily, increasing the signal-to-noise ratio (SNR) of the pulse wave signal. To prevent damage to the piezoelectric ceramics, a layer of film between the piezoelectric ceramics and the measuring surface is required. This film is composed of high-density fibers, which can prevent the permeation of sweat while also allowing pulse waves to penetrate. The piezoelectric ceramic we use is the same material as the piezoelectric ceramic used in the buzzer, making the sensor we designed easy to manufacture. The synthetic materials of this type of piezoelectric ceramic are zinc oxide (ZnO), lithium niobate (LiNbO_3_), lithium tantalite (LiTaO_3_), and barium titanate (BaTiO_3_).

Figure 2 illustrates the change of the piezoelectric sensor when the pulse wave transmits in the radial artery. The volume change Δ*V* when the pulse wave transmits to the sensing period and the skin movement is calculated per Equation (1). Equation (2) indicates that the internal pressure change Δ*P* of the pressurized sensor is the sum of the pressure *P_A_* before being pressurized and the pulse wave pressure *P_a_*. Therefore, the proportional relationship between the pressure changes of *P_A_* and Δ*P* is determined in Equation (3), which involves the ratio of specific heats γ, which is about 1.4 for air [17]. The mass *m* is then *SL* times the density of air *ρ,* as per Equation (4), moved by the difference in pressure between the artery and piezoelectric ceramic. Newton’s law of acceleration is Equation (5). We can find the relationship between the force *F* and the action force of the piezoelectric ceramic with Equation (6).
(1)ΔV=V−S⋅L
(2)$$\Delta P=P_{A}+P_{a}$$
(3)$$\frac{P_{a}}{P_{A}}=-\gamma \frac{\Delta V}{V}=-\gamma \frac{S\cdot L}{V}$$
(4)m=S⋅L⋅ρ
(5)$$\frac{F}{m}=\frac{d^{2}x}{dt^{2}}$$
(6)$$\frac{d^{2}x}{dt^{2}}=\frac{P_{a}\cdot S}{S\cdot L\cdot \rho }=-\frac{\gamma \cdot S\cdot P_{A}}{\rho \cdot V\cdot L}\cdot L$$
where *V* is the piezoelectric sensor’s volume before the pulse wave is pressurized, *S* is the section area, and *L* is the displacement length of the skin movement of the pulse.

Figure 3 shows the position where the pulse wave sensor is placed on the wearable device. This study uses two sensors to measure the subject’s pulse wave and pulse transit time (PTT) [18]; then, the distance between the two sensors divided by the PTT can be converted to a PWV. The wearable device with the dual-piezoelectric sensor can detect and filter out the pulse wave when an arrhythmia occurs. The MK equation is applied based on hemodynamics to determine the blood pressure with this PWV. This approach avoids the issue of blood pressure measurement errors in patients with arrhythmia using the oscillometric method.

### 2.2. Analog Front End Design

The voltage signal generated by piezoelectric ceramics is on the millivolt scale. In this study, the analog front end (AFE) is designed to amplify the voltage signal of piezoelectric ceramics and use filters to improve the SNR. Figure 4 shows the processing of piezoelectric signals from piezoelectric ceramics by various amplifiers of the AFE. Figure 5a shows the voltage follower (VF), which is the infinite input impedance, providing the complete input of the piezoelectric ceramic signal on the mV level. If we connect the piezoelectric sensor to the external load, we get a high-pass filter. The cut-off frequency of the high-pass filter is the point in the circuit with a 3dB drop in magnitude. Therefore, we need to add an external load resistor to ensure the quality of the piezo signal. To determine the corresponding load resistor *R*1 in Equation (7), we need to measure the parasitic capacitance *C_p_* of the piezoelectric ceramics [19], and *f* is the cut-off frequency of 1 Hz of heartbeats with rates around 60 beats per second. Table 1 shows the piezoelectric sensor specifications, and the *C_p_* of the piezoelectric ceramics is 8000 pF. It can be calculated that the most suitable load resistor is about 20 M ohms.

Figure 5b is a signal amplifier, and the gain is adjusted by selecting *R*2 and *R*3 through Equation (8). The signal passes through the signal amplifier, and the voltage signal of the pulse wave is amplified to hundreds of millivolts. When the voltage signal of the pulse wave is amplified, the noise will also be amplified. Figure 6a is the second-order Sallen–Key high-pass filter (HPF); Equation (9) is its transform function calculation formula, and the cut-off frequency is determined by Equation (10). The cut-off frequency of the HPF set in this study is 0.58 Hz; *R*4 and *R*5 are equal, and *C*4 and *C*5 are also designed with the same value. After the signal passes through the HPF, the DC component can be filtered out.

Figure 6b is the second-order Sallen–Key low-pass filters (LPF) [20]; Equation (11) is its transform function calculation formula, and the cut-off frequency is determined by Equation (12). In this study, the cut-off frequency of the LPF is 10.6 Hz; *R*6 and *R*7 are equal, as are *C*6 and *C*7, which can attenuate the interference caused by the supply mains frequency and other disturbances.
(7)$$R1=\frac{1}{\left(2\cdot \pi \cdot f\cdot C_{p}\right)}$$(8)$$Vout=Vin\cdot \frac{R2}{R3}$$(9)$$\frac{Vout(s)}{Vin(s)}=\frac{s^{2}}{s^{2}+s\left(\frac{1}{R4\cdot C4}+\frac{1}{R4\cdot C5}\right)+\frac{1}{R4\cdot C4\cdot R5\cdot C5}}$$(10)$$fc_{HPF}=\frac{1}{2\pi \sqrt{R4\cdot C4\cdot R5\cdot C5}}$$(11)$$\frac{Vout(s)}{Vin(s)}=\frac{\frac{1}{R6\cdot C6\cdot R7\cdot C7}}{s^{2}+s\left(\frac{1}{R7\cdot C6}+\frac{1}{R6\cdot C6}\right)+\frac{1}{R6\cdot C6\cdot R7\cdot C7}}$$(12)$$fc_{LPF}=\frac{1}{2\pi \sqrt{R6\cdot C6\cdot R7\cdot C7}}$$

## 3. Piezoelectric Sensor Signal Processing

A 12-bit analog-to-digital converter (ADC) samples the output voltage of the AFE at a rate of 5 kHz. The pulse wave signal measured by the sensor converts the mechanical wave to a digital signal. This conversion process introduces high-frequency noise. Therefore, this research designs a finite impulse response (FIR) band-pass filter for the pulse wave signal sampled by the ADC. To detect the position of the pulse wave, we designed an algorithm that uses the amplitude of the sensor signal to adjust the gradient dynamically. The dynamic gradient adjustment can better adapt to individuals with different pulse strengths, and a dynamic gradient detection algorithm can find the position of normal sinus rhythm and arrhythmia.

### 3.1. Digital Filter Design

In this study, FIR is used to filter the signal after ADC samplings [21]. As shown in Equation (13), the pulse wave signal *x* sampled by ADC and the filter coefficient *b_k_* are convolutional, obtaining the filtered pulse wave signal *y*. As in Equation (14), the filter coefficient needs to calculate the frequency response and window function coefficients according to different FIR filter types and convolution kernel size. We use a band-pass filter (Equation (15)) and set the cut-off frequencies *f_c_*_1_ and *f_c_*_2_ to 0.7 Hz and 9.5 Hz, respectively. This range covers the pulse wave signal composition, where *ω_c_*_1_ and *ω_c_*_2_ are the cut-off frequencies of the band-pass filters, and *f_c_*_1_ and f_c2_ are the transition frequencies, which need to be divided by the ADC sampling rate of 5 kHz (Equation (16)). The term *w* is the window function of the Kaiser window [22], as defined in Equation (17), and *N* is the window size, chosen as 128 in this study. The variable *I*_0_ is the modified Bessel function of the second kind, as defined in Equation (18), where *β* can adjust the frequency domain’s main lobe and sidelobe levels.
(13)$$y[n]=\sum_{k=0}^{N-1}b_{k}\cdot x[n-k]$$
(14)$$b[n]=w[n]\cdot h_{d}[n],0\leq n\leq N-1$$
(15)$$h_{d}[n]=\left\{\begin{array}{c} \frac{\sin(\omega_{c2}(n-N))}{\pi (n-N)}-\frac{\sin(\omega_{c1}(n-N))}{\pi (n-N)},n\neq N\\ \frac{\omega_{c2}-\omega_{c1}}{\pi },n=\frac{N}{2} \end{array}\right.$$
(16)$$\omega_{c1}=\frac{f_{c1}}{5000},\omega_{c2}=\frac{f_{c2}}{5000}$$
(17)$$w[n]=w_{0}\left(\frac{L}{N}(n-\frac{N}{2})\right)=\frac{I_{0}[\beta \sqrt{1-{\left(\frac{2n}{N}-1\right)}^{2}}]}{I_{0}[\beta ]},0\leq n\leq N$$
(18)$$I_{0}=\sum_{i=1}^{n}{\left(\frac{{\left(\beta /2\right)}^{i}}{i!}\right)}^{2}$$

### 3.2. Adaptive Threshold and Peak Detection

The pulse wave amplitudes of different subjects can vary. Therefore, the threshold of the detected pulse wave signal must be dynamically adjusted. The response speed of the dynamic threshold is dependent on the calculated size. In this study, we use half of the sampling rate as the calculated size for the dynamic threshold.

Equation (19) is the calculation method of the dynamic threshold, and Equation (20) is the formula for covariance (*CV*), which is used to calculate the rate of change of the past signal strength. Before calculating the *CV*, the mean value of the signal amplitude is determined per Equation (21). The root mean square (*RMS*) is the amplitude level of the signal per Equation (22). The *gain* can adjust the level of the dynamic threshold, and we chose one in this study. When the SNR is lower, the waveform of the pulse wave has a smaller *CV* for the overall signal. Increasing the gain can reduce the false positive of peak detection.
(19)$$\mathrm{gradient}=\mathrm{RMS}\cdot \frac{CV}{100}\cdot gain$$
(20)$$CV=\frac{{\left(\sum_{k=0}^{N-1}x_{k}-mean\right)}^{2}}{N}$$
(21)$$mean=\frac{\sum_{k=0}^{N-1}x_{k}}{N}$$
(22)$$RMS=\sqrt{\frac{\sum_{k=0}^{N-1}x_{k}^{2}}{N}}$$

Figure 7 shows the pseudocode [23] of the peak detection algorithm proposed in this study. The *gradient* is the dynamic threshold calculated by Equation (19), the index is the order of sampling points, and *value* is the current signal amplitude. When the signal amplitude starts to rise, the pulse wave is transmitted to the sensor. When the current amplitude is greater than the previous value, the local maxima are updated; the signal amplitude is less than the local maximum minus the gradient, and the previous local maximum is the peak of the pulse wave. The local minimum is updated when the signal amplitude decreases and the current amplitude is smaller than the previous value. When the signal amplitude is greater than the local minimum plus the gradient, the local minimum is the trough of the pulse wave. This algorithm only needs to update the gradient and record local maxima and minima for real-time pulse wave detection. It is suitable for running on wearable devices with limited memory and performance.

Figure 8 shows the result of the algorithm used in this study. The four pulse wave signals in the figure include three normal sinus rhythms and one arrhythmia. The yellow star indicates that the pulse wave is rising when the signal strength is greater than the last minimal plus gradient. The red circle is the peak of the pulse wave. The green triangle is when the signal amplitude is less than the last maxima minus the gradient, indicating the pulse wave is falling. The black circle is the trough of the pulse wave.

When the peak position of each pulse wave can be determined, we can count the peak-to-peak time difference of a set of measurement sequences. The algorithm identifies the pulse waves whose time difference is much larger or smaller than the average value, effectively detecting the pulse when an arrhythmia occurs. Placing pulse wave sensors at two points in the artery can measure two pulse waves that are the same but with a phase shift. After the peaks of the two pulse waves are detected, the phase shift time difference between the two is calculated as the PTT. The PTT can be converted into a PWV through the known distance of two points of the sensor.

## 4. Result and Analysis

### 4.1. Experiment Equipment and Setup

Figure 9a shows the pulse wave measurement device that integrates two piezoelectric sensors. The distance between the two sensors is 2.5 cm. The device was placed on the subject’s wrist and transmitted the pulse wave signal to USB data acquisition software (Figure 9b), which sampled the piezoelectric sensor signal at a rate of 5 kHz and ran the pulse wave detection algorithm proposed in Section 3 in real time to detect the position of the normal sinus rhythm and the arrhythmia. Furthermore, the data from the acquisition software were stored in the file for analysis.

To verify that the piezoelectric sensor could receive the pulse wave when an arrhythmia occurs, the subject’s ECG signal was recorded simultaneously as a reference. Figure 9c is the ECG measurement device (ADI AD8232 IC [24]) and limb lead (MLII). The chip is an AFE circuit designed for ECG measurement. The AD8232 could sense the ECG signal of the human body, and sample the ECG signal at 360 Hz through a 12-bit ADC, and transmit it to the ECG signal acquisition software (Figure 9d) through a USB. The ECG signal acquisition software recorded the subject’s signal as a reference for verifying the occurrence of arrhythmia by the piezoelectric sensor. This study performed pulse wave and ECG measurements simultaneously, and the recording time was one minute.

Figure 10a,b shows blood pressure measurement devices and wrist cuffs based on the oscillometric method. The Wheatstone bridge pressure sensor in this study (MPS-3117-006GC) is the type of sensor used by typical electronic blood pressure monitors. The pressure curve of the device was calibrated with the mercury manometer. Our electronic sphygmomanometer air pump, with a range of 40 to 180 mmHg, pressurized the wrist cuff at an increased speed of 4 mmHg per second in closed-loop control and was connected to the data acquisition software (Figure 10c) through a USB. At a sample rate of 1 kHz, the subject’s envelope waveform was recorded. A subject with arrhythmia measured by the oscillometric method would have distortion [25].

To avoid the issues with the oscillometric method measuring arrhythmia in subjects—the distorted waveform and the blood pressure estimation error—we used a pulse wave measuring device that integrates two piezoelectric sensors. The PWV of the subject could be measured, and the MK equation based on hemodynamics estimated blood pressure. Since the MK equation requires the subject’s arterial diameter and tube wall thickness, an ultrasound imaging device was used in this study to sample the radial artery image of the subject’s wrist. Figure 11 shows the ultrasound imaging device Sonostar U-Probe-L6C transducer, which has 196 elements. We sampled the subject’s radial artery and took the average artery diameter and wall thickness [26] to verify the MK equation. The Korotkoff sound is the gold standard [27] for non-invasive blood pressure measurement. The subject wore a cuff on the arm with a stethoscope placed between the cuff and arm. The researcher then pressurized the cuff and listened for the brachial artery pulse sound through the stethoscope. After no pulse sound was heard, the researcher would leak air at 2 mmHg per second. The pressure point of the first pulse heard during the decrease of the cuff pressure is the systolic blood pressure (SBP), and the last pulse is the pressure point of the diastolic blood pressure (DBP). Figure 12a shows the Welch Allyn 5098-27 DS66 aneroid sphygmomanometer used in this study, and Figure 12b shows the cuff and the subject.

### 4.2. Protocol

This study acquired the pulse wave signals of arrhythmia from patients with PAC/PVC. The experimental protocol is shown in Figure 13 and described in the following. The subject needs to first rest for 5 min before the test. Then, the researcher secures the subject’s wrist to prevent movement. The ECG limb lead II and pulse wave measurement device are attached to the subject’s wrist, and the ECG and pulse wave signal are acquired for one minute. The signal sequence is obtained three times in total, with the subject resting for one minute after each interval. The researcher then sets up the blood pressure measurement device and cuff on the subject’s wrist. The cuff position is the same as that of the pulse wave measurement device. The blood pressure measurement device acquires the envelope waveform ranging from 40 to 180 mmHg. Finally, the researcher uses an aneroid sphygmomanometer to sample the reference blood pressure of the subject, and an ultrasound imaging device to sample the subject’s radial artery images.

### 4.3. Pulse Wave Signal Analysis of Atrial Fibrillation

Figure 14 shows the ECG and pulse wave signal sequence of a PAC patient. When another area of the atrium depolarizes before the sinus node and thus triggers a premature heartbeat [28], the ECG signal indicates that PAC occurs, making irregular changes in the heart rhythm. The pulse wave signal during a PAC can be detected by the algorithm proposed in this research to detect the irregular position of the pulse wave.

Figure 15 shows the ECG and pulse wave signal sequence of a PVC patient. The characteristic of PVC in the ECG signal is that the QRS complex is long and deformed [29]. The pulse wave signal responds to the pressure of the heart when the heart ejects changes when PVC occurs—the hemodynamics in the arteries change, and the amplitude of the first peak of the pulse wave signal is significantly attenuated to close to the second peak. The algorithm proposed in this study can dynamically adjust the threshold to detect the position of the first peak.

Figure 16 shows the oscillometric envelope waveform of a PAC/PVC patient. When a pressurized electronic sphygmomanometer is used to measure the blood pressure of an arrhythmia patient during an arrhythmia, the pulse in the envelope waveform disappears, and the envelope waveform distorts. The oscillometric method uses the pulse with the maxima amplitude as a reference, then finds the pressure point of a specific ratio from the amplitude of each pulse during the measurement, as the subject’s SBP and DBP. Figure 16 clearly shows the issue of the oscillometric method measuring arrhythmia patients being unable to estimate blood pressure.

### 4.4. Results

In this study, we refer to the QRS complex position to label the location of the pulse wave signal. The pulse wave signal has a first peak and a second peak. We use the local maxima of the first peak of the pulse wave as the reference point to label. For the local maxima labeling point of each pulse wave, the position of the error within the three points detected by the algorithm is a true positive (TP). If the position detected by the algorithm is out of the range of three points, or if it is not detected, it is a false negative (FN). This study’s pulse wave sampling rate was 5 kHz, so the error of three points is approximately 0.6 ms. Table 2 shows the analysis results of the pulse wave signals of PAC patients. The total beat (TB) of the normal pulse wave was 187, of which TP was 186 and FN was 1; the sensitivity (Se) and the accuracy (Ac) were 99.4%, and the pulse wave when PAC occurred had a TB of 19, TP of 17, FN of 2, Se of 89.4%, and Ac of 89.4%. Figure 17 compares the PWVs of PAC patients, including the pulse waves of PACs and the statistical distribution of PWV with the PAC removed (normal pulse waves). The range of values significantly reduced without PAC, so the accuracy of PWV can be improved if the pulse wave of PAC can be detected and filtered.

Table 3 lists the results of analyzing the pulse wave signals of PVC patients. The TB of the normal pulse wave was 195, while TP was 195, FN was 0, Se was 100.0%, and Ac was 100.0%. The pulse wave when PVC occurred had a TB of 14, TP of 12, FN of 2, Se of 85.7%, and Ac of 85.7%. We used the same signal analysis as the previous section. The amplitude of the first peak of the pulse wave when PVC occurred was significantly attenuated when too close to the second peak. The algorithm detection is more complex than with the pulse wave signal when PAC occurs. Figure 18 compares the PWVs of patients with PVC, including the pulse waves of PVCs, the statistical distribution, and the normal pulse wave after removing the PVC. The results are similar to Figure 17. The range when PVC was included was more extensive than the statistical distribution of PWV with PVC filtered out.

Equation (23) is the MK equation, where *E_inc_* is Young’s modulus, *D* is the diameter of the radial artery (m), *h* is the wall thickness of the radial artery (m), and *ρ* is the whole blood density of 1061 kg/m^3^ [30]. Equation (24) illustrates the relationship between PWV and pressure *P* (mmHg), where *E_inc_* is Young’s modulus, and *E*_0_ and γ are used as coefficients in this study, and are 1428.7 and 0.031, respectively [31]. The Young’s modulus of the previous article is from pressure-volume curves of the aorta in animal experiments [32]. In the relationship between PWV and *P* of the MK equation, previous studies have shown that *P* is approximately the mean arterial pressure (MAP) because the systolic period is 1/3 and the diastolic period is 2/3 [33]. Therefore, MAP is approximately equal to 1/3 of SBP plus 2/3 of DBP. For comparison, we convert the SBP and DBP measured by the aneroid sphygmomanometer into MAP through Equation (25).
(23)$$PWV=\sqrt{\frac{E_{inc}\cdot h}{D\cdot \rho }}$$
(24)$$E_{inc}=E_{0}\cdot \exp^{\xi \cdot P}$$
(25)$$MAP=\frac{SBP+(2\cdot DBP)}{3}$$

Table 4 shows the physiological parameters of the PAC/PVC subjects. We used the MK equation to estimate the MAP of each PWV and then averaged the MAPs. The PWVs of the pulse wave signals when an arrhythmia occurred are not excluded. The mean difference of the MAP estimate by the MK equation and the aneroid sphygmomanometer was greater than the mean difference of MAP that filters the pulse wave signal during an arrhythmia. This is because when an arrhythmia occurs, the hemodynamics of the radial artery change, making the pulse wave unstable and causing an error in the calculation of PWV. Therefore, estimating blood pressure by PWV with the MK equation can avoid the issue of the oscillometric method. Electing pulse wave signals when an arrhythmia occurs and filtering out arrhythmia can further improve its accuracy.

## 5. Conclusions and Future Work

The piezoelectric sensor proposed in this study measured the pulse wave signal when a subject’s arrhythmia occurred. We verified that the piezoelectric ceramic with a cavity structure and our AFE design could measure the occurrence of PAC/PVCs. The pulse wave detection algorithm was verified based on the dynamic threshold of our proposed technique, which can recognize ectopic beats. We used a piezoelectric sensor to collect the pulse wave signals of patients with arrhythmia, showing that the algorithm detects the pulse wave’s position when the PAC/PVC in the arrhythmia occurs. Our experimental results prove that our proposed pulse wave detection method has an accuracy of 99.4% for PAC patients’ normal pulse wave signal and 89.4% for the pulse wave when PAC occurs. The normal pulse wave signal for PVC patients has an accuracy rate of 100%, and the pulse wave when PVC occurs has an accuracy rate of 85.7%. We compare the mean difference between the MAP of the pulse wave when the arrhythmia occurs and the aneroid sphygmomanometer. For PAC patients, the MAP result of the pulse wave without PAC is 4.76 mmHg, and the MAP with the pulse wave when PAC occurs is 2.84 mmHg. For PVC patients, the MAP result of the pulse wave without PVC is 7.7 mmHg, and the MAP that filters out the pulse wave when PAC occurs is 4.4 mmHg. Therefore, for PAC/PVCs patients, filtering out the pulse wave when an arrhythmia occurs can effectively reduce the error with the aneroid sphygmomanometer and significantly improve the accuracy of non-invasive blood pressure measurement for arrhythmia patients. This approach also avoids the oscillometric method’s inaccuracy caused by the distortion of the envelope curve during an arrhythmia. Future research can use this method to obtain more pulse waves of different types of arrhythmia patients through experiments. The method we propose provides non-invasive blood pressure measurement technology during an arrhythmia that improves the accuracy of the wearable device for patients with arrhythmia.

## Figures and Tables

**Figure 1 sensors-21-06915-f001:**
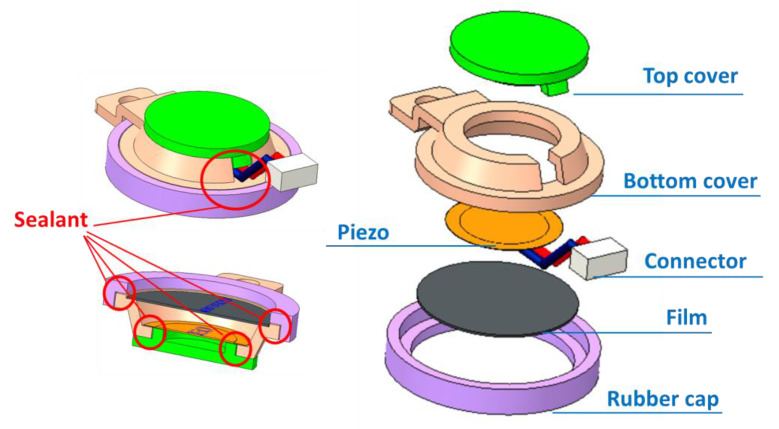
The cavity structure of the pulse wave sensor.

**Figure 2 sensors-21-06915-f002:**
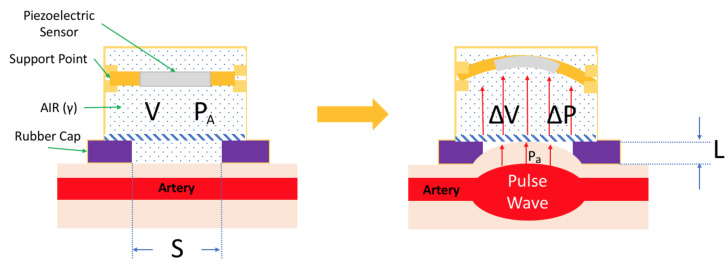
Change of the piezoelectric sensor in cavity housing when the pulse wave transmits in the radial artery.

**Figure 3 sensors-21-06915-f003:**
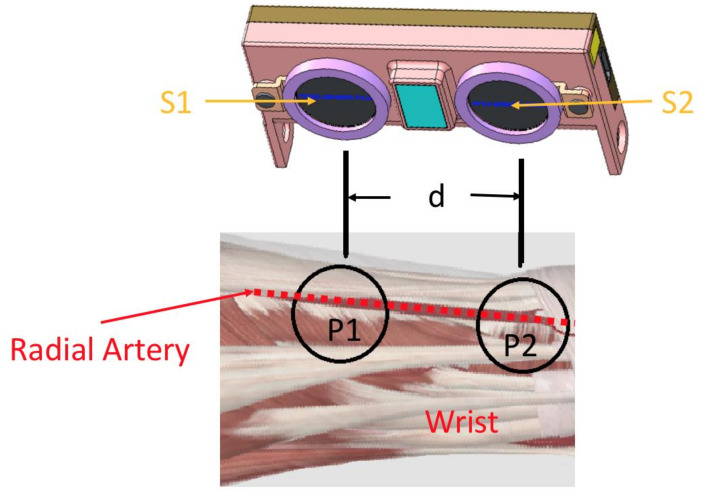
Position where the pulse wave sensor is placed on the wearable device.

**Figure 4 sensors-21-06915-f004:**
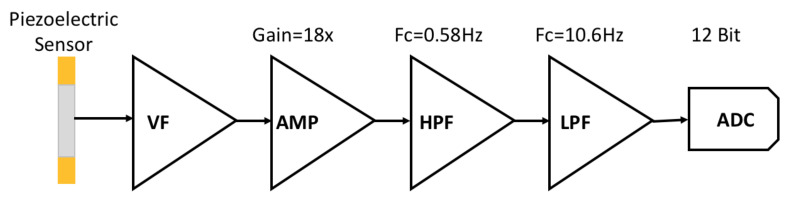
Processing of piezoelectric signals from piezoelectric ceramics by various amplifiers of the AFE.

**Figure 5 sensors-21-06915-f005:**
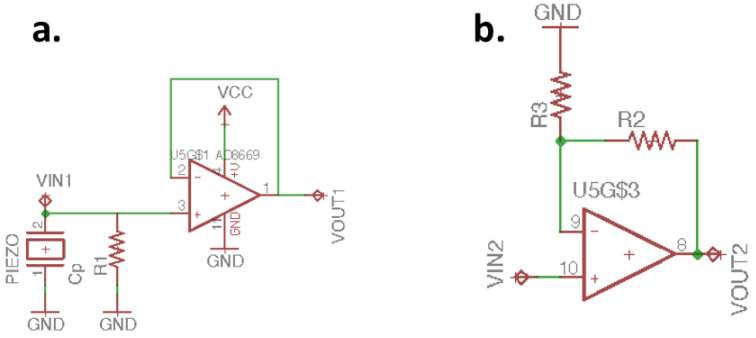
(**a**) VF and (**b**) signal amplifier and gain adjust.

**Figure 6 sensors-21-06915-f006:**
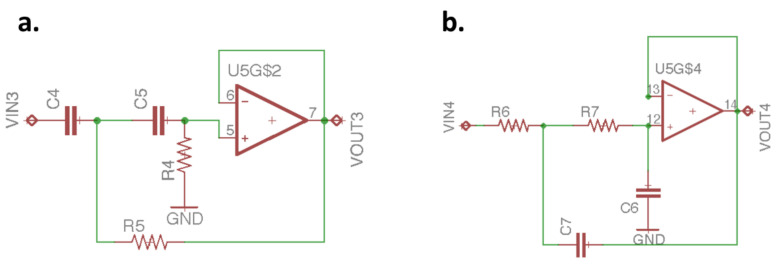
(**a**) Second-order Sallen–Key HPF and (**b**) second-order Sallen–Key LPF.

**Figure 7 sensors-21-06915-f007:**
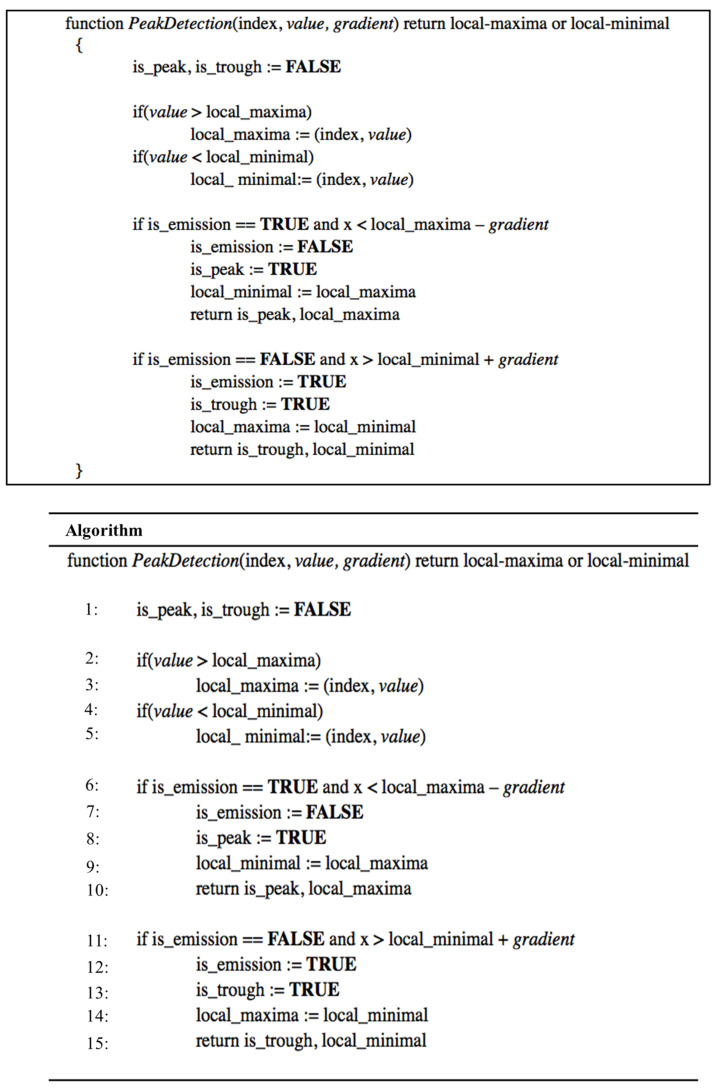
Pseudocode of the peak detection algorithm.

**Figure 8 sensors-21-06915-f008:**
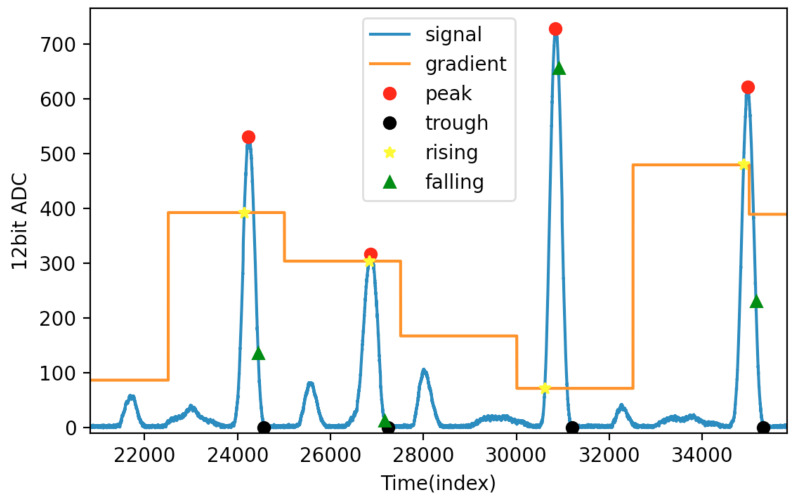
Result of the algorithm used in this study.

**Figure 9 sensors-21-06915-f009:**
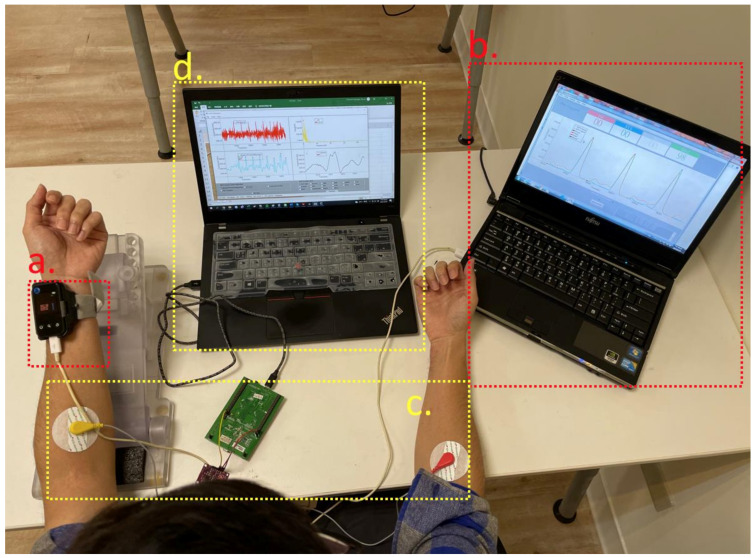
(**a**) Pulse wave measurement device. (**b**) USB data acquisition system. (**c**) ECG measurement device and limb lead. (**d**) ECG signal acquisition software.

**Figure 10 sensors-21-06915-f010:**
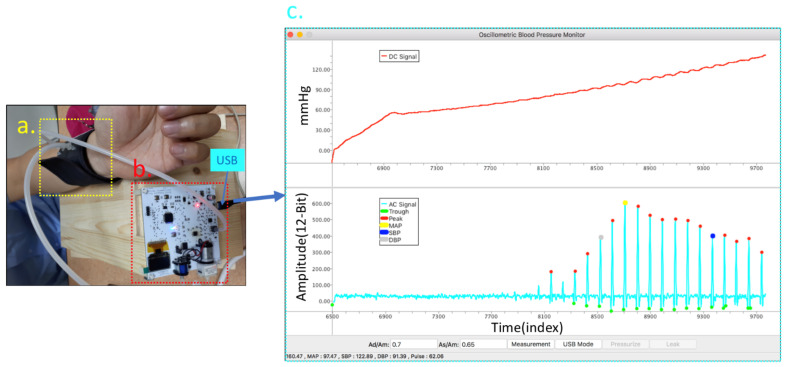
Blood pressure measurement device: (**a**) wrist cuffs based on the principle of the oscillometric method, (**b**) Wheatstone bridge pressure sensor, (**c**) electronic sphygmomanometer air pump.

**Figure 11 sensors-21-06915-f011:**
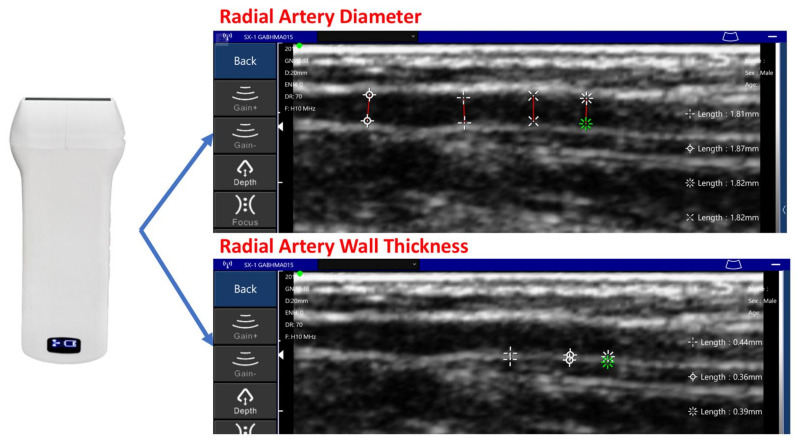
Ultrasound imaging device Sonostar U-Probe-L6C transducer.

**Figure 12 sensors-21-06915-f012:**
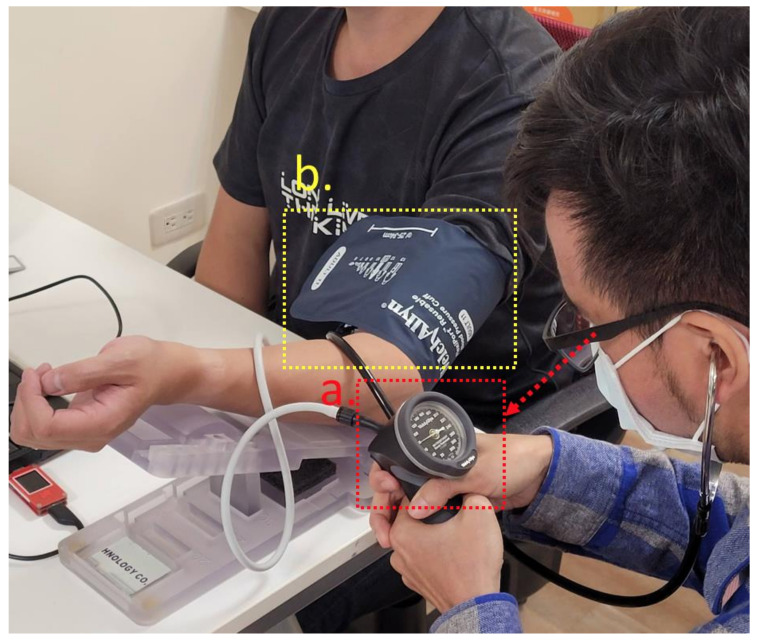
(**a**) Welch Allyn 5098-27 DS66 aneroid sphygmomanometer; (**b**) subject wearing the cuff with a stethoscope between the cuff and the arm, and the researcher pressurizing the cuff.

**Figure 13 sensors-21-06915-f013:**
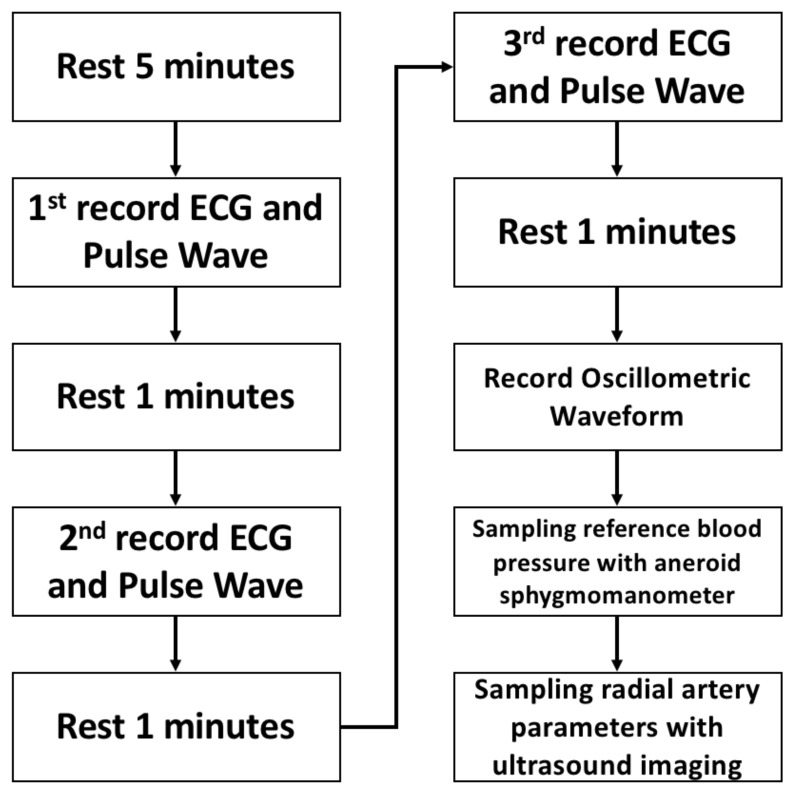
Experimental protocol.

**Figure 14 sensors-21-06915-f014:**
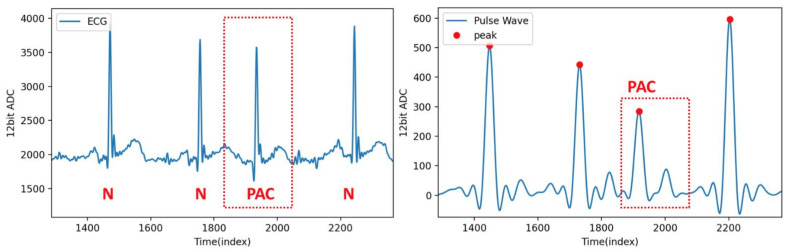
ECG and pulse wave signal sequence of a PAC patient.

**Figure 15 sensors-21-06915-f015:**
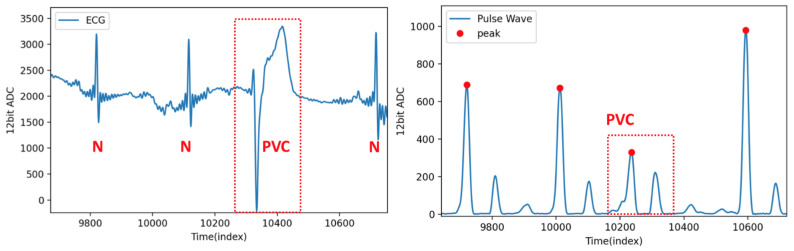
ECG and pulse wave signal sequence of a PVC patient.

**Figure 16 sensors-21-06915-f016:**
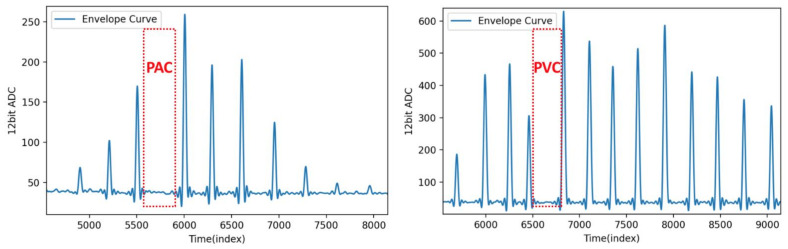
The oscillometric envelope waveform of a PAC/PVC patient.

**Figure 17 sensors-21-06915-f017:**
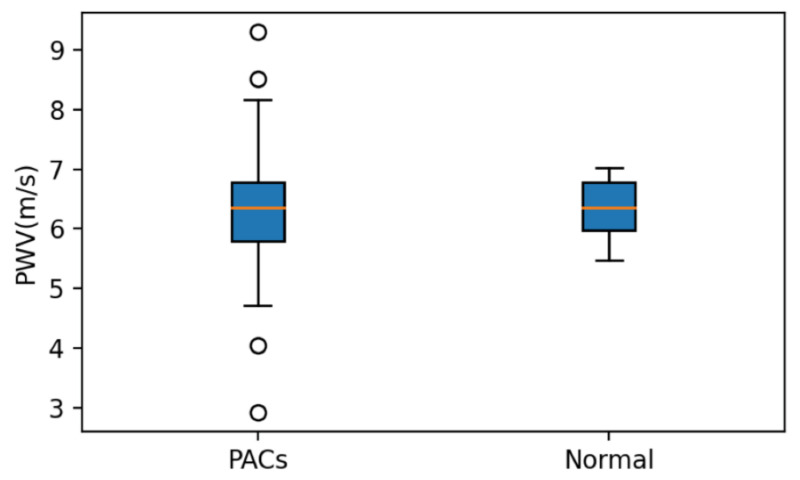
PWVs of PACs compared to the statistical distribution of PWV with the PAC removed (normal pulse wave).

**Figure 18 sensors-21-06915-f018:**
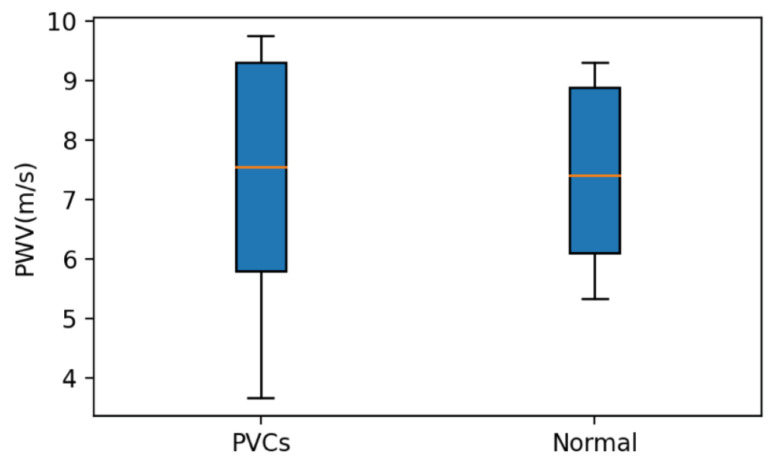
PWVs of PVC and the statistical distribution of the PWV of the normal pulse wave after removing the PVC.

**Table 1 sensors-21-06915-t001:** Piezoelectric sensor specifications.

Specification	Detail
Capacitance (pF, at 120 Hz)	8000 ± 30%
Plate material	Brass
Plate material (mm)	Φ10 × 0.05
Ceramic size (mm)	Φ7 × 0.07
Weight (g)	0.2
Price (USD)	0.35

**Table 2 sensors-21-06915-t002:** Results of analyzing the pulse wave signal of PAC patients.

Pulse Wave Statistic	TB	TP	FN	Se (%)	Ac (%)
Normal	187	186	1	99.4	99.4
Abnormal	19	17	2	89.4	89.4

**Table 3 sensors-21-06915-t003:** Results of analyzing the pulse wave signals of PVC patients.

Pulse Wave Statistic	TB	TP	FN	Se (%)	Ac (%)
Normal	195	195	0	100	100
Abnormal	14	12	2	85.7	85.7

**Table 4 sensors-21-06915-t004:** Physiological parameters of the PAC/PVC subjects.

Subject Characteristics	Patient with PACs	Patient with PVCs
Artery diameter (mm)	1.75	2.35
Artery wall thickness (mm)	0.39	0.47
Mean PWV_Arrhythmia_ (m/s)	6.15	6.77
Mean PWV_Normal_ (m/s)	6.34	7.14
SBP_Reference_ (mmHg)	112	126
DBP_Reference_ (mmHg)	74	86
MAP_Reference_ (mmHg)	86.54	99.2
Mean MAP_Arrhythmia_ (mmHg)	81.78	91.5
Mean MAP_Normal_ (mmHg)	83.7	94.8
MD_Reference−Arrhythmia_ (mmHg)	4.76	7.7
MD_Reference-Normal_ (mmHg)	2.84	4.4

## Data Availability

Not applicable.

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
