# Peer review of "Piezoelectric Sensor for the Monitoring of Arterial Pulse Wave: Detection of Arrhythmia Occurring in PAC/PVC Patients"

_sensors, 2021, doi:10.3390/s21206915_

Round 1

Reviewer 1 Report

The authors proposed a high-sensitivity pulse sensor that can measure the hemodynamic characteristics of the pulse wave and then estimate the blood pressure even if an arrhythmia occurs. The sensor consists of a piezoceramic plate and a cavity structure. Accompanying electronic parts and a signal processing algorithm were also developed. Efficacy of the sensor was illustrated by applying the sensor to measure the blood pressure of a patient who had the arrhythmia.

This manuscript actually consists of two parts, i.e. (1) development of a piezoelectric sensor shown in Fig. 1 and accompanying electronic parts, and (2) use of the piezoelectric sensor to measure the blood pressure of a patient with an arrhythmia utilizing the Moens–Korteweg equation.

The second part looks fine. It differed from the conventional oscillometric method and could estimate the blood pressure in a non-invasive way during an arrhythmia. However, the first part does not seem to have any novelty.

The authors claim that they proposed a piezoelectric sensor that can accurately measure the blood pressure of a patient with an arrhythmia. If they did the measurement just to demonstrate the efficacy of the proposed sensor, the main subject is the new sensor. However, the sensor in Fig. 1 does not look new. It is just an encapsulation of two piezoceramic plates. Equations (1)-(3) are not sufficient to explain the operation principle of the sensor.

Further, it is not clear why the authors could not use conventional piezoelectric pressure sensors and why they had to develop the structure in Fig. 1. There are numerous commercial piezoelectric pressure sensors available on the market. Review of existing sensors for blood pressure measurement is missing in the manuscript. Just using two PVDF films available on the market seems to be good enough to replace the sensor in Fig. 3. The electronics including the VF, OP amp, HPF, LPF, and ADC are not new at all.

Therefore, the manuscript needs to be revised to clarify the novelty and imperativeness of the new piezoelectric sensor. If the imperativeness of the new sensor is not appropriate, the manuscript needs to be revised to emphasize only the second part, i.e. new measurement technique in comparison with existing methods.

One more minor comment:

  • In Fig. 2, ‘L’ is used to denote the rubber cap thickness. In Fig. 3, the same symbol ‘L’ is used to denote the distance between the two piezoelectric sensors. One of the two L’s should be replaced with another symbol to avoid confusion.

Reviewer 2 Report

In my opinion, the manuscript is suitable for publication in Sensors journal but Authors must complete a major revision. Manuscript should be revised according to following comments:

1. Chapter “Introduction” must be improved:
a) Authors should present a literature review in the range of piezoelectric sensors used for the blood pressure measurement. There is no literature review in this field in the manuscript, despite  the fact that piezoelectric sensors are applied for monitoring of arterial pulse wave. Author should justify the need of presented research with state of art,
b) Authors should describe what is new in proposed piezoelectric sensor in comparison to other piezoelectric sensors which are used for monitoring of arterial pulse wave,

2. Chapter “Materials and design” must be improved:
a) line 72 and others: Authors write that “piezoelectric ceramics” is part of sensor, but there is no description of used piezoceramic ceramic. Many types of piezoceramics are known nowadays. Piezoelectric material must be more precisely described because is a subject of manuscript,
b) Figure 2: on the basis of description for this figure, can be noticed that the piezoelectric sensor contains only the piezoelectric material (sensor does not include the rest of the elements, e.g. support point?). Description for this figure should be corrected,
c) line 113: Author should explain term “parasitic capacitance”,
d) line 114: Authors should explain why the range of frequency is from 5 to 12 kHz,
e) Equation 4: this equation described optimal load resistance for which the electric power is the largest. Why such value of load resistance was selected?

3. Chapter “Result and Analysis” must be improved:
a) Figure 10: quality of this figure should be improved,
b) line 351: authors should describe the kind of material for which Young’s modulus are given in line 354, 

4. Chapter “Conclusion and future work” must be improved:
a) this chapter must be substantially processed, because remind abstract. Conclusions should be scientific statements on the basis of obtained results of experiments. Conclusions should fill research gap which is described in introduction. 

Reviewer 3 Report

The author developed a compact, inexpensive, and highly sensitive pulse wave sensor to detect arrhythmia in a non-invasive manner. The sensor is designed based on piezoelectric ceramics, which converts the deformation generated by pressure of the pulse wave to voltage signals. The mechanical design, signal analysis, experimental setup, and blood pressure measurement demo are well demonstrated in the manuscript. Overall, this study is thorough and well demonstrated. The manuscript is well-written and easy to follow. Thus, I believe the manuscript is worth publishing on Sensors.

Round 2

Reviewer 1 Report

The revised manuscript addressed all the points I raised, so I recommend the revised manuscript for publication in its present form. 

Reviewer 2 Report

Accept in present form.